# Genetically Engineered Lung Cancer Cells for Analyzing Epithelial–Mesenchymal Transition

**DOI:** 10.3390/cells8121644

**Published:** 2019-12-15

**Authors:** Michał Kiełbus, Jakub Czapiński, Joanna Kałafut, Justyna Woś, Andrzej Stepulak, Adolfo Rivero-Müller

**Affiliations:** 1Department of Biochemistry and Molecular Biology, Medical University of Lublin, 20-093 Lublin, Poland; michalkielbus@umlub.pl (M.K.); jakub.czapinski@umlub.pl (J.C.); joanna.kalafut@umlub.pl (J.K.); andrzejstepulak@umlub.pl (A.S.); 2Postgraduate School of Molecular Medicine, 02-091 Warsaw, Poland; 3Department of Clinical Immunology, Medical University of Lublin, 20-093 Lublin, Poland; justyna.wos@umlub.pl; 4Faculty of Natural Sciences and Technology, Åbo Akademi University, 20500 Turku, Finland

**Keywords:** epithelial–mesenchymal transition (EMT), mesenchymal–epithelial transition (MET), cancer cell line, vimentin, reporter

## Abstract

Cell plasticity, defined as the ability to undergo phenotypical transformation in a reversible manner, is a physiological process that also exerts important roles in disease progression. Two forms of cellular plasticity are epithelial–mesenchymal transition (EMT) and its inverse process, mesenchymal–epithelial transition (MET). These processes have been correlated to the poor outcome of different types of neoplasias as well as drug resistance development. Since EMT/MET are transitional processes, we generated and validated a reporter cell line. Specifically, a far-red fluorescent protein was knocked-in in-frame with the mesenchymal gene marker *VIMENTIN* (*VIM*) in H2170 lung cancer cells. The vimentin reporter cells (VRCs) are a reliable model for studying EMT and MET showing cellular plasticity upon a series of stimulations. These cells are a robust platform to dissect the molecular mechanisms of these processes, and for drug discovery in vitro and in vivo in the future.

## 1. Introduction

The ability of cells to temporally acquire different characteristics, also known as cell plasticity, plays essential roles in physiological and pathophysiological processes [1]. Epithelial cells might transform to a mesenchymal phenotype and return to epithelial phenotype by epithelial–mesenchymal transition (EMT) and mesenchymal–epithelial transition (MET), respectively. EMT has been correlated to the metastasis and invasion potential of many types of malignancies [2]. The fact that the poor outcome of many types of neoplasias correlates with EMT/MET, makes these molecular phenomena an important focus for research and drug targeting [3,4,5,6]. Despite the clinical association, the role of EMT/MET in metastasis is inconclusive; for example, mesenchymal-like prostate cancer cells survive in circulation, but, unlike epithelial or cells undergoing EMT, they are unable to form macrometastases [7]. In fact, breast cancer cell metastasis to lung tissue in mice was not affected by decreasing EMT—by targeting EMT—triggering transcription factors such as *SNAI1* (*Snail*) and *SNAI2* (*Slug*) by overexpressing miR-200 [8]. An additional role of EMT in tumorigenesis might be the development of apoptotic tolerance and increased resistance to chemotherapy as has been found in animal models and patients [8,9]. Recently, a hybrid stage between the epithelial and mesenchymal phenotypes (hybE/M) has been recognized, and such hybE/M cells migrate outside the primary tumors, displaying some mesenchymal features such as spindle-like morphology, increased nuclear levels of ZEB1 transcription factor, and epithelial properties such as cell–cell adhesion potential [10,11].

In order to study the transitory stages of EMT, MET, and hybE/M in vivo, there is need to generate reporter cells to visualize phenotypical changes. There have been different approaches to this, for example, the use of heterologous fluorescent proteins driven under stage-specific promoters such as mesenchymal (*ZEB1*) and epithelial (*CDH1*) [12], mesenchymal *snai1* and epithelial *sox10* in zebrafish [13], or mesenchymal *Vimentin (Vim)* in mice [14]. Nevertheless, heterologous expression of exogenous reporter genes is hampered by the existence of alternative promoters, *cis*- and *trans*-regulatory elements, and epigenetic events that modulate promoter activation, making some data unreliable.

To overcome these limitations, we genetically engineered and characterized human lung carcinoma H2170 Vimentin Reporter Cells (VRCs) where the fluorescent protein coding gene (*mCardinal*) has been knocked-in as a genetic fusion, although as separate proteins due to a T2A self-cleaving peptide, at both alleles of the mesenchymal marker *VIM*.

## 2. Materials and Methods

### 2.1. Cell Culture, Nucleofection, and Transfection

Human squamous lung carcinoma (H2170), human colorectal adenocarcinoma (HT29), and human embryonic kidney (HEK293) cells were cultured according to standard mammalian tissue culture protocols and sterile techniques. The cell lines were cultured in DMEM supplemented with 10% fetal calf serum (FCS), 100 units/mL and penicillin/100 µg/mL streptomycin. All tissue culture media and supplements were obtained from Gibco. H2170, HT29, and HEK293 cells were obtained from ATCC. H2170 cells were nucleofected for genome editing with the use of Nucleofector I Device and Cell Line Nucleofector Kit T (Amaxa). The optimized protocol is as follows: nulceofection of 1 million suspended cells with 2 µg of the plasmid DNA using program X-001, generating a transfection efficiency of 99%. For performing functional experiments on a smaller scale, the H2170 cells were transfected with the use of Lipofectamine 3000 (Invitrogen, Waltham, MA, USA) with transfection efficiency 90–95% after 24 h. A total of 50000 cells were seeded a day before transfection in 24-well plates. Transfection was performed using 500 ng of the plasmid DNA, 1.5 μL Lipofectamine^®^3000 Reagent, and 1 μL P3000™ Reagent (both from Invitrogen) per well, following the manufacturer’s protocol. HEK293 were transfected with the use of TurboFect reagent (Thermo, Waltham, MA, USA) according to the protocol supplied by the company. Next day, the transfection efficiency was in the range of 80–90%. VRCs were also cultured with 10 ng/mL TGFβ (human TGFβ1, Biorbyt, Cambridge, United Kingdom) for 72 h.

### 2.2. Plasmid Vectors for the Knock-In of VIM

We modified Cas9/gRNA-expressing plasmid pSpCas9(BB)-2A-Puro (PX459) V2.0 (Addgene plasmid #62988) by inserting the fragment coding the targeting gRNA using digestion/ligation protocol [15]. The oligonucleotides used to generate the gRNA were gRNA-F and gRNA-R, as seen in Appendix A. The template plasmid used for inserting DYKDDDDK-tagged (FLAG) mCardinal fluorescent protein gene after *VIM* was designed as following (gRNAsite—800 nt of *Vim*—P2A—mCardinal—FLAG—800 nt of 3′*Vim*UTR—gRNAsite) and was synthetized by Thermo. The Cas9-gRNA and the template plasmids were both nucleofected to cells at the same time, and the cells were selected 2 days after nucleofection and cultured for another 2 days in 5 μg/mL of puromycin. The positive single cell clones obtained by dilution were genotyped, as described in the Genotyping section.

### 2.3. Plasmid Vectors Used in Functional Assays

For functional experiments in H2170 knocked-in cells, we used (FLAG) Snail 6SA (active Snail) plasmid, which was a gift from Mien-Chie Hung (Addgene plasmid # 16221) [16], TGFB1-bio-His (proTGFβ), which was a gift from Gavin Wright (Addgene plasmid # 52185) [17], and HA-OVOL2 (OVOL2)-expressing plasmid, which was a gift from Changwon Park [18]. EMT/MET in VRCs was studied with the use of expressing vectors harboring genomic fragments of the microRNA-200 family (*miR-145*, *miR-200b*, *miR-200c*, *miR-205*) which were cloned in our laboratory (See the Molecular Cloning section). Control cells were transfected with pUC18 subcloning plasmid.

### 2.4. Molecular Cloning

All the plasmid fragments used for cloning were amplified using tiHybrid proofreading DNA polymerase (EURx), according to the supplied protocol. PCR products amplified on the plasmid DNA template were incubated overnight at 37 °C with DpnI FastDigest enzyme (Thermo) as per the manufacturer’s instructions.

The *VIM-T2A-mCardinal* sequence was cloned from cDNA of knocked-in cells upon PCR amplification and linearized by PCR using a pmR-expressing vector (Clonetech) and recombined using Gibson Assembly (NEB). The resulting vector contained a full *VIM-T2A-mCardinal* reading frame under the control of a CMV promoter. The primers for insert amplification were KI-F and KI-R, whereas the pair used for backbone linearization were BCB-F and BCB-R, as seen in Appendix A. Mutagenesis was performed by REPLACR methodology [19], using the SDM-F and SDM-R primers, as seen in Appendix A.

The vectors harboring *miR-145*, *miR-200b*, *miR-200c*, and *miR-205* genomic fragments were created by inserting each PCR-amplified microRNA gene into the 3′UTR of mNeon-expressing vector (pmR-mNeon). All genomic fragments listed above were amplified using tiHybrid DNA polymerase (EURx) from DNA, which was purified from the blood of healthy volunteer with the use of GeneAll Exgene Blood SV kit (GeneAll). The sets of primers used for amplification of the *miR-145*, *miR-200b*, *miR-200c*, and *miR-205* fragments were *miR145-F*/*miR145-R*, *miR-200b-F*/*miR-200b-R*, *miR-200c-F*/*miR-200c-R*, *miR-205-F*/*miR-205-R*, respectively, as seen in Appendix A. The amplified products produced sticky ends upon digestion by BglII and HindIII restrictases (both from Thermo). Digested and purified DNA fragments were ligated using T7 ligase (Thermo) in molar ratio 3:1 with 100 ng of linear pmR-mNeon, which was previously cut by BglII and HindIII enzymes.

The resulting vectors were named *miR-145*, *miR-200b*, *miR-200c*, and *miR-205*. The sequences of all the vectors were verified by Sanger sequencing (Genomed, Warsaw, Poland).

### 2.5. Genotyping

The targeting sequence for CRISPR/Cas9 was in the last intron (intron 8) of *VIM* in HEK293 and H2170 cells with the use of Benchling algorithm. Single-cell clones were cultured on 96-well plates to more than 50% confluence (Nunc, Roskilde, Denmark). Upon washing with phosphate-buffered saline (PBS, Gibco), they were genotyped by PCR using Mouse Direct PCR Kit (Bimake), following the manufacturer’s instructions. The primers used for genotyping were 170 and 249, as seen in Appendix A).

The genotyping was further confirmed by PCR using the same set of primers (170 and 249), tiHybrid DNA polymerase and high-quality genomic DNA, purified from single-cell clones with the use of QIAamp DNA Mini Kit (Qiagen, Hilden, Germany). The KI was verified by sequencing (Genomed).

### 2.6. RNA Extraction, Reverse Transcription, and qPCR

Total RNA was isolated from cells with the use of Extractme Total RNA kit (Blirt, Gdansk, Poland) according to the manufacturer’s manual, including DNase treatment. The purity and quantity of isolated RNA was estimated spectrophotometrically with the use of a Tecan M200Pro microplate reader supplied with NanoQuant plates (Tecan, Zürich, Switzerland). Only the samples with 260/280 nm OD ratio higher than 1.8 were used for downstream analysis. For molecular cloning, 3 μg of RNA were reverse transcribed for 30 min at 50 °C using an oligo(dT) primer and the Transcriptor High Fidelity cDNA Synthesis Kit (Roche, Rotkreuz, Switzerland) followed by 5 min enzyme inactivation at 85 °C, according to the manufacturer’s instructions. For QPCR, 2 μg of RNA were reverse transcribed using a High-Capacity RNA-to-cDNA Kit (Applied Biosystems, Waltham, MA, USA) following to the manufacturer’s protocol.

Quantitative real-time expression analysis was performed using a LightCycler^®^480 II instrument (Roche) equipped with 384 well plates and PowerUp™ SYBR™ Green Master Mix (Applied Biosystems). The primers were as listed in Appendix A, as seen in Appendix A. Amplification was performed in 12.5 μL reaction mixture containing cDNA amount corresponding to 12.5 ng of total RNA, 1 × PowerUp SYBR Green Master Mix, and 3.125 pmol of each primer (forward and reverse).

After 2 min of initial incubation at 50 °C followed by 2 min incubation at 95 °C, cDNA was amplified in 45 cycles consisting of 15 s denaturation at 95 °C, 30 s annealing at 60 °C and 20 s elongation at 72 °C. The obtained fluorescence data was analyzed using a relative quantification (RQ) method 2^–ΔΔCT^ for estimating expression fold changes normalized to dim-VRCs and 2^–ΔCT^ method for comparison of the expression of each measured gene. The assessed genes expression (*VIM*, *mCardinal*, *CDH1*, *ZEB1*, and *ZEB2*) were normalized to *GAPDH* level, which was measured with the use of GAPDH-F and GAPDH-R oligonucleotides. *GAPDH* was previously confirmed as stably expressed at the mRNA level in H2170 cells as well as in VRCs (mean Cp = 17.14; median Cp = 17.09; SD = 0.5209; SEM = 0.03638; N = 205). Expression of *VIM* was measured using VIM-F and VIM-R primers, measurement of *mCardinal* level was conducted using mCard-F and mCard-R primers, whereas estimation of *CDH1* expression was performed at the mRNA level with the use of Cdh1-F and Cdh1 -R oligonucleotides. *ZEB1* and *ZEB2* quantifications were performed using ZEB1F/R and ZEB2F/R pairs of oligonucleotides, respectively. *TWIST1* and *TWIST2* quantifications were performed using the TWIST1F/R and TWIST2F/R oligonucleotide pairs, respectively. The primers were designed as intron-spanning to avoid any influence of genomic DNA contamination and are listed in Appendix A.

### 2.7. Flow Cytometry of Living Cells

VRCs or HT29 cells cultured for one day on a six-well plate (Nunc) were washed with HBSS (Hank′s Balanced Salt Solution, Thermo) and detached by Accutase (Corning, Corning, NY, USA). The cells dissolved in 100 µL HBSS with 5 µL of PE Mouse anti-E-Cadherin (562526, BD Pharmingen, San Jose, CA, USA) were incubated for 1 h, washed with HBSS, and suspended in 0.5 mL HBSS. The cells were counted on the FL-3 fluorescence channel of a FACSCalibur flow cytometer (BD).

### 2.8. Cell Sorting

VRCs seeded one day before were then sorted using a BD FACSAria™ flow cytometer (BD Biosciences, San Jose, CA, USA). In this case, cells were selected into two populations due to the intensity of their mCardinal fluorescence in the far-red channel, as seen in Appendix A. After sorting, the purity of the sorted cells was confirmed by flow cytometry and reached more than 97%. The resulting two populations of VRCs were named dim-VRCs and bright-VRCs.

### 2.9. Confocal Imaging of Living Cells

The cells were seeded onto 24-well glass-bottom plates (MoBiTec, Goettingen Germany) a day before transfection. Transfection was conducted as described above. Transfected cells were then visualized under a Nikon Ti Confocal microscope after 24 and 28 h since transfection, using a 563 nm laser for mCardinal fluorescence. Mean fluorescence intensity was measured using NISelements (ver 3.22.08, Melville, NY, USA) software and is shown on the graphs. Each of at least 10 measured regions of interest (ROIs) included a cluster of more than 20 adherent cells. The brightest slices of the z-stacks were chosen for measurement. The measurements were done in duplicate.

### 2.10. Immunocytochemistry

The cells seeded a day before on glass bottom Labtec chamber slides (Nunc) were washed with PBS and fixed with 2% paraformaldehyde for 20 min at room temperature. Upon washing in PBS, the cells were permeabilized by 0.5% triton in PBS for 20 min. The endogenous peroxidase was blocked by incubation in 1% sodium azide and 1% hydrogen peroxide in PBS for another 20 min. Another washing step in PBS was followed by incubation in Blocking Buffer which was 1× Normal Donkey Serum (reconstituted from 20×, Jackson ImmunoResearch, Ely, United Kingdom) containing 1% bovine serum albumin (SantaCruz Biotechnology, Dallas, TX, USA) and 0.1% of triton for 30 min in room temperature. Supplied by Cell Signaling, primary antibodies against VIM (rabbit, #5714S Danvers, MA, USA) and FLAG-tag (mouse, #8146S) were diluted in Blocking Buffer at ratios of 1:100 and 1:1600, respectively. Incubation with the primary antibodies was conducted overnight at 4 °C and was followed by washing in PBS. Secondary Peroxidase F(ab’)₂ Fragment Donkey Anti-Rabbit IgG (H+L) (711-036-152, Jackson ImmunoResearch) for detection of primary rabbit and Peroxidase F(ab’)₂ Fragment Donkey Anti-Mouse IgG (H+L) (715-036-150, Jackson ImmunoResearch) for detection of primary mouse antibodies were diluted 1:1000 in Blocking Buffer. Incubation with secondary antibodies (fluorochrome or peroxidase conjugated) was performed for 1 h at room temperature and was followed by washing in PBS. Peroxidase-conjugated antibodies were stained with Alexa Fluor 555 Tyramide Reagent (B40955, LifeTechnolgies, Carlsbad, CA, USA) or Alexa Fluor 488 Tyramide Reagent (B40953, LifeTechnolgies) following to the manufacturer’s protocol, resulting in enzyme-linked fluorescence signal amplification. When double Alexa Fluor Tyramide Reagents (488 and 555) staining was done, an additional step of 15 min washing in 3% hydrogen peroxide, 0.1% sodium azide in PBS, to achieve complete inhibition of HRP, was added between incubations with two different secondary HRP-antibodies. The cells were finally washed by PBS and stained with Hoechst (Cayman) diluted in PBS (1:1000 from 10 mg/mL stock solution). The cells were washed again, mounted with ProLong Gold mounting medium (LifeTechnologies), and visualized under a Nikon Ti confocal microscope.

### 2.11. Migration of VRCs

A XCELLigence real-time cell analysis system (Acea) equipped with 16-well CIM-plates served for the VRCs migration assessment. VRCs were first detached using Accutase, then transfected with OVOL2, *miR-200c*, *miR-205*, or pUC18 plasmids using Lipofectamine300 reagent, as described above. Then, 30000 cells were seeded on each well of the CIM-plate, which was kept in the standard mammalian tissue culture conditions and measured over 24 h. The cells’ migration toward an attractant, which was 10% DMEM supplemented with 10% FCS, was normalized to cell migration toward DMEM alone and shown as Cell Index. Each experiment was measured at least in triplicate and repeated twice.

## 3. Results

### 3.1. Genome Editing

The targeting sequence in H2170 cells was confirmed by sequencing, as seen in Appendix A. The H2170 lung cancer cells were nucleofected with CRISPR and template (vKIT), as seen in Figure 1A, plasmids, following enrichment using puromycin. vKIT contained the sequence of the self-cleaving T2A peptide followed by the FLAG-tagged *mCardinal* fluorescent protein gene, flanked by two 800 nucleotide homology arms (HA) that correspond to *VIM* gene fragments near the CRISPR targeting region, as seen in Appendix A.

The template cassette, containing *HA-T2A-FLAG-mCardinal-HA*, was flanked by two gRNA targeting sequences identical to those for the *VIM* gene in order to linearize the knock-in insert and promote efficient homology-directed repair (HDR).

Two days after nucleofection, the cells were selected using puromycin (1 µg/mL) for another two days. Single cell clones were obtained by dilution, further they were genotyped by PCR with efficiency near 2.7% (3/112). Single cell clones, confirmed by PCR, were called VRCs and used in downstream analysis. The sequence of the knocked-in DNA fragment in the VRCs was further verified by Sanger sequencing, as seen in Appendix A.

The subcellular localization of mCardinal in living cells, as seen in Figure 1B–D, by confocal imaging showed that mCardinal was present in the cytoplasm, but its concentration was not homogenous, showing bright foci as well as cytoplasmic localization. Immunofluorescent labelling of VIM and FLAG-mCardinal, as seen in Figure 2A–F, shows almost complete colocalization, where VIM was found in the cytoskeleton as well as in the foci.

The unexpected cellular localization of VIM led us to analyze if the gene product was intact. For this, we amplified the entire *VIM-T2A-mCardinal* sequence from mRNA and cloned it into an expression vector. The whole *VIM-T2A-mCardinal* sequence was sequence-verified. This VIM-T2A-mCardinal cDNA exerted identical cellular distribution of mCardinal under the confocal microscope when expressed in HEK293 cells, as seen in Appendix A.

To determine if fusing mCardinal to VIM would result in a better cellular distribution, the T2A peptide sequence was modified by substitution of the P16A and P18A in the T2A sequence. The resulting vector (*VIM–mCardinal*) was overexpressed in HEK293 cells, showing granular localization too, as seen in Appendix A. Overexpression of VIM–mCardinal in HEK293 cells showed that this fusion protein localized only in large granules in the cell and is similar to the fluorescence in VIM–T2A–mCardinal overexpressing cells, where the fluorescence was shown in similar granules as well as diffusing into the cytoplasm, as seen in Appendix A.

Since the foci in the VIM–T2A–mCardinal cells corresponded to fused VIM–mCardinal fusions, likely resulting from inefficient T2A processing, they were brighter than if evenly distributed throughout the cytoplasm; therefore, we considered them as a baseline.

To further characterize VRCs, they were sorted according to fluorescence intensity, resulting in two populations (dim-VRCs and bright-VRCs) that differed by about three-fold in mCardinal fluorescence intensity, as seen in Appendix A. The difference was reduced upon culturing to less than two-fold mCardinal intensity between dim-VRCs and bright-VRCs after 48 h, as seen in Figure 2G. *VIM* and *mCardinal* expression measured at the transcriptional level confirmed that those two genes are expressed at equal levels, as seen in Figure 2H and Appendix A.

The dim-VRCs where then exposed to a series of EMT-inducing factors to analyze the correlation between mesenchymal conversion and mCardinal expression. At the same time VRCs were exposed to TGFβ (10 ng/mL) for 72 h to evaluate the expression levels of EMT-related genes, including: *CDH1*, *VIM*, *SNAI1*, *ZEB1*, *ZEB2*, *TWIST1*, and *TWIST2*, as seen in Figure 3A. We observed that the transcripts of *VIM*, *SNAI1*, *ZEB1*, *ZEB2*, and *TWIST2* were significantly elevated in TGFβ-treated VRCs in comparison to untreated VRCs. We have not observed significant changes in *CDH1* or *TWIST1* expression, although there is a tendency for repression and transcriptional activation, respectively. In order to determine if TGFβ may increase migratory capacity of VRCs, we conducted migration assay were we compared migration kinetics of VRCs treated with TGFβ to untreated VRCs. We observed that that migratory potential of the cells treated with TGFβ was elevated in comparison to those untreated (Figure 3B). We also examined the mCardinal localization in VRCs upon treatment with TGFβ via confocal microscopy, as seen in Figure 3C–H. Confocal imaging did not show any changes in localization of mCardinal in VRCs upon TGFβ stimulation in comparison to untreated cells.

### 3.2. EMT Markers vs. Reporter Gene

The dim-VRCs were transfected with active *Snai1* or proTGFβ-expressing plasmids, which resulted in a two-fold increase in fluorescence, examined using confocal microscopy 24 h or 48 h after transfection, as seen in Figure 4A. Correspondingly, the transcriptional levels of *VIM* and *mCardinal* showed similar fold increase (RQ), as seen in Figure 4B,C. The highest expression of *VIM* was seen in the cells that overexpressed active Snai1 one and two days after transfection (median RQ = 1.625 and RQ = 1.580, respectively). proTGFβ-overexpressing dim-VRCs after 24 h and 48 h also overexpressed *VIM* at median levels; RQ = 1.371 and RQ = 1.424, respectively. The control population of bright-VRCs expressed *VIM* at median RQ = 1.569. The highest level of *mCardinal* was expressed in active Snai1-expressed dim-VRCs after 24 h (median RQ = 1.357) and 48 h after transfection (median RQ = 1.240). proTGFβ-overexpressing dim-VRCs also overexpressed mCardinal at median levels RQ = 1.206 and RQ = 1.248 after 24 h and 48 h, respectively, whereas bright-VRCs expressed *mCardinal* at a median RQ of 1.181. The expression levels of *VIM* and *mCardinal* relative to *GAPDH* showed a similar pattern, as seen in Figure 4B,C. The epithelial marker *CDH1* in proTGFβ-overexpressing dim-VRCs did not reveal significant changes 24 h or 48 h after transfection, as seen in Figure 4D. We did not observe significant changes in *CDH1*, *VIM*, *SNAI1*, *ZEB1*, *ZEB2*, *TWIST1*, or *TWIST2* levels between VRCs and parental H2170 cells by qPCR, as seen in Appendix A.

### 3.3. OVOL2 and miRNAs Overexpression Modulates CDH1 Expression in VRCs via ZEB1/2 Repressors

In order to test if mesenchymal-like VRCs could reverse their phenotype into a more epithelial state, we transfected these cells with a strong epithelial activator, OVOL2 [20]. Immunostaining followed by flow cytometry analysis and qPCR confirmed our hypothesis, showing that the E-cadherin level is elevated in OVOL2-overexpressing VRCs after 48 h, as seen in Figure 4E. OVOL2-overexpressing VRCs concomitantly showed a significant increase in mCardinal fluorescence, as seen in Figure 4F. To examine the mechanism of epithelial phenotype development by VRCs, we analyzed the expression level of CDH1, as well as its negative regulators *ZEB1* and *ZEB2*, as seen in Figure 5A–C, in the cells that overexpressed OVOL2 or four different *ZEB1/2*-targeting members of the *miR-200* family (*miR-145*, *miR-200b*, *miR-200c*, *miR-205*). Both negative regulators of *CDH1* expression (*ZEB1* and *ZEB2*) were significantly downregulated in the VRCs that overexpressed *OVOL2* or *miR-205* with simultaneous upregulation of *CDH1*. *miR-200c* overexpression in VRCs produced increased level of *CDH1* together with downregulated ZEB1, whereas the effect of *miR-145* overexpression was seen only as an elevated *CDH1* level. We did not observe changes in *CDH1* or *ZEB1/2* levels in *mirR-200b-*expressing VRCs.

### 3.4. OVOL2 and miR-200c or miR-205-Overexpressing VRCs Show Decreased Migratory Potential

In order to examine if decreased levels of ZEB1/2 could repress the migratory properties of VRCs [20,21], the cells were transfected with *OVOL2*, *miR-200c*, and *miR-205*-overexpressing plasmids and a real-time migration assay was conducted. We observed an over two-fold decrease in the number of migrating VRCs that overexpressed all of the tested factors that may modulate *ZEB1/2* (*OVOL2*, *miR-200c*, and *miR-205*) in comparison to the mock transfected cells, as seen in Figure 5D.

## 4. Discussion

Mounting evidence shows that tumors are far more heterogenous than expected in due to genetic diversity of the tumor cells as well as their phenotypic plasticity [22,23]. This phenotypic plasticity is in fact phenotype switching, which is understood as a phenomenon whereby cancer cells transition between different phenotypes in response to environmental cues, without acquiring new mutations [24]. This cellular plasticity has been reported as having high clinical impact because it is crucial for drug resistance development, e.g., in lung cancer patients treated with EGFR inhibitors [25], reacquiring pluripotency and become a cancer stem cell (CSC) [26], or maintaining metastatic ability of many types of cancer cells [27,28,29].

Among tumor cells, there are some that undergo EMT, MET, or hybE/M, and in order to understand the effects of cellular plasticity in biological and pathological processes, there is a need for reporters that can determine the stage of a cell. Observing changes in mesenchymal and epithelial phenotype as they occur is a significant improvement in studying molecular mechanisms in cancer cells. H2170 cells were chosen because they have been a good model for EMT and MET [30,31].

To date, EMT/MET is routinely studied by the use of exogenous reporter genes that are heterologously expressed, exhibiting interference of *cis-* and *trans*-regulatory elements, alternative promoters, or epigenetic events, which result in obtaining unreliable data [32,33]. These limitations are overcome by using genome engineering of endogenous genes. There are a few of available cancer cell lines that harbor a C-terminal red fluorescent protein (RFP) tag on VIM: A549 (lung), HCT116 (colorectal), and MDA-MB-231 (breast adenocarcinoma), and they are commercially available at the ATCC cell repository. These cells, however, have *RFP* knocked-in to the beginning of last exon of *VIM* (exon 9), which results in deletion of a large fragment of that exon and of the gene product (https://www.lgcstandards-atcc.org/en/Global/Products/CCL-247EMT.aspx#documentation). While these modified A549, HCT116, and MDA-MB-231 cell lines enable near real-time tracking of the EMT/MET status as cells transition from epithelial to mesenchymal phenotype under defined conditions [34], any approach where an endogenous protein is truncated may affect functions in comparison to endogenously expressed proteins, particularly in the context of a protein such as VIM, which has many signaling modulatory activities during cell plasticity [35,36,37].

We used a far-red fluorescent protein (mCardinal), which is simultaneously expressed with endogenous VIM, but separate during translation due to the viral self-cleaving peptide (T2A). T2A was chosen because, together with P2A, it has been reported as the most efficient of all the tested self-cleaving 2A peptides [38,39,40,41]; moreover, T2A resulted in the least amount of “uncleaved” protein product among the family of 2A peptides [39,40,41]. The stability and half-life of the two resulting proteins can result in small changes in total expression, the two proteins are synthesized at a 1:1 ratio [41]. *VIM* expression at the transcriptional level corresponded to that of *mCardinal*, and when stimulated, the changes of *VIM* expression were hand-to-hand with those of *mCardinal*, evidencing that our strategy is fully functional.

mCardinal was chosen mainly because this far-red monomeric protein is better for in vivo settings due to it is excitation as wavelengths above 600 nm, which penetrate through hemoglobin-rich tissues far better than lower wavelengths [42]. The second reason is that mCardinal is suitable for precise monitoring of its changing expression due to its short maturation half-time (27 min) [43], which is about three times shorter compared to other commonly used fluorescent markers, e.g., RFP [44].

The subcellular localization of mCardinal in VRCs by confocal imaging show nonhomogenous distribution of the fluorescence, which was observed at moderate levels in the cytoplasm and in brighter glowing foci. Immunofluorescent staining of VIM and FLAG-tag mCardinal showed colocalization at the “foci”, whereas Vimentin was also present in the cytoskeleton of VRCs. Moreover, recloning of the *VIM–P2A–mCardinal* and further overexpressing it in HEK293 cells shown similar subcellular distribution of the fluorescence. The presence of foci is probably the result of inefficient T2A peptide cleavage, with the fusion protein being unable to form normal VIM polymers [39,40,41], which is more conspicuous in the case of the fusion of VIM–mCardinal. In fact, C-tags in VIM might result in an impaired protein [45], and its expression may give a dominant-negative phenotype, similar to the phenotype observed in the cells which express only the N-terminal domain of VIM (NTD-VIM) [46]. NTD-VIM (VIM that lacks C-terminus) domain-expressing cells show its abnormal localization, as formation of the granules in the cytoplasm together with disrupted endogenous VIM localization examined by immunostaining [45]. The role of the tail domain in VIM organization is not fully understood, although it is known to be necessary for appropriate network formation [47]. Thus, it seems to be possible that abnormal VIM localization in VRCs may be produced by the fluorescent protein. Yet, the presence of bright foci is advantageous as these structures are brighter than diffused cytoplasmic mCardinal under low VIM expression.

Dim-VRCs presented a reduced mesenchymal phenotype compared to bright-VRCs. The dim-VRCs could be driven to a more mesenchymal phenotype via expression of proTGFβ [17] or constitutively active SNAI1 transcription factor [16]. We also noticed insignificant downregulation of *CDH1* in proTGFβ overexpressing dim-VRCs, what suggests that E-cadherin may be only partially regulated via the TGFβ-dependent pathway [48].

*OVOL2*-overexpressing VRCs respond by E-cadherin expression, which is in accordance with data describing OVOL2 as a strong epithelial regulator that maintains transcriptional programs in epidermal keratinocytes and mammary epithelial cells by repressing ZEB1 and ZEB2 [49,50].

The MiR-200 family was identified as double-negative regulators by silencing *CDH1* and transcription repressors *ZEB11* and *ZEB2* [51,52]. We confirmed the activatory role of *miR-145*, *miR-200c*, and *miR-205* on *CDH1* expression in VRCs. We also confirmed that *miR-200c* expression decreased *ZEB1* levels in VRCs [53], but we did not observe any effect of *miR-200c* on *ZEB2* expression [54]. VRCs that overexpressed *miR-205* showed *ZEB1/2* downregulation, which is in accordance with previous works [55,56]. Our results suggest that the molecular mechanisms of the miR-200 family on EMT might be cell-type-specific, and more work needs to be done to clarify this. Our data is in accordance with previous data in that the migratory potential of mesenchymal cells is inhibited by ZEB1 or ZEB2 downregulation produced by miR-200 family overexpression [57].

The very similar levels of *CDH1*, *VIM*, *SNAI1*, *ZEB1*, *ZEB2*, *TWIST1*, and *TWIST2* transcripts between VRCs and the parental H2170 cell line suggests that VRCs may be useful in broad range of studies.

In conclusion, our data uniquely illustrate a reporter line, VRCs, as a reliable reporter model for studying EMT and MET. VRCs allow one to directly observe cellular plasticity with respect to their mesenchymal/epithelial state in vitro and, in the future, in vivo. These cells could also be used as a robust platform for drug development.

## Figures and Tables

**Figure 1 cells-08-01644-f001:**
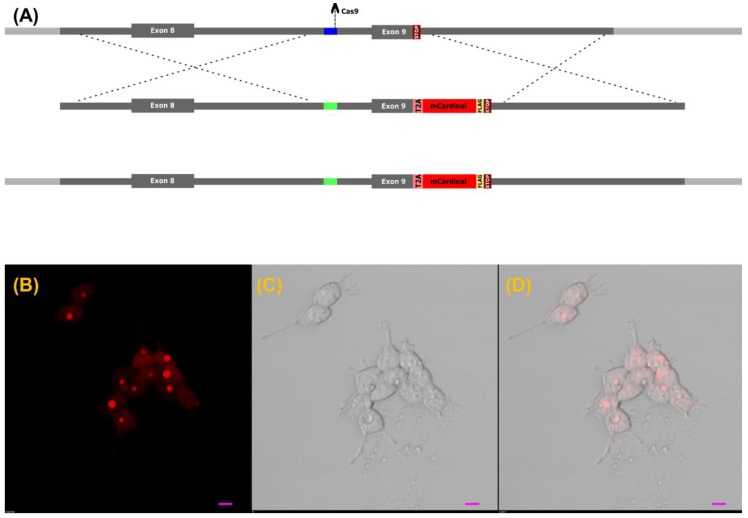
(**A**) Schematic representation of the genome editing strategy. Genomic region of H2170 cells targeted by CRISPR/Cas9 with homology arms marked in grey, while the targeted site in intron 8 is marked in dark blue. The donor DNA template contains the sequence of T2A-mCardinal-FLAG, flanked by two homology arms that correspond to VIM gene fragments (grey). The VIM allele in VRCs contains the knocked-in DNA, and the altered sequence recognized by Cas9 is colored in green. (**B**–**D**). Subcellular localization of mCardinal fluorescent protein in living VRCs. The scale bar represents 10 µm. (**B**) mCardinal, (**C**) transmission, (**D**) merged.

**Figure 2 cells-08-01644-f002:**
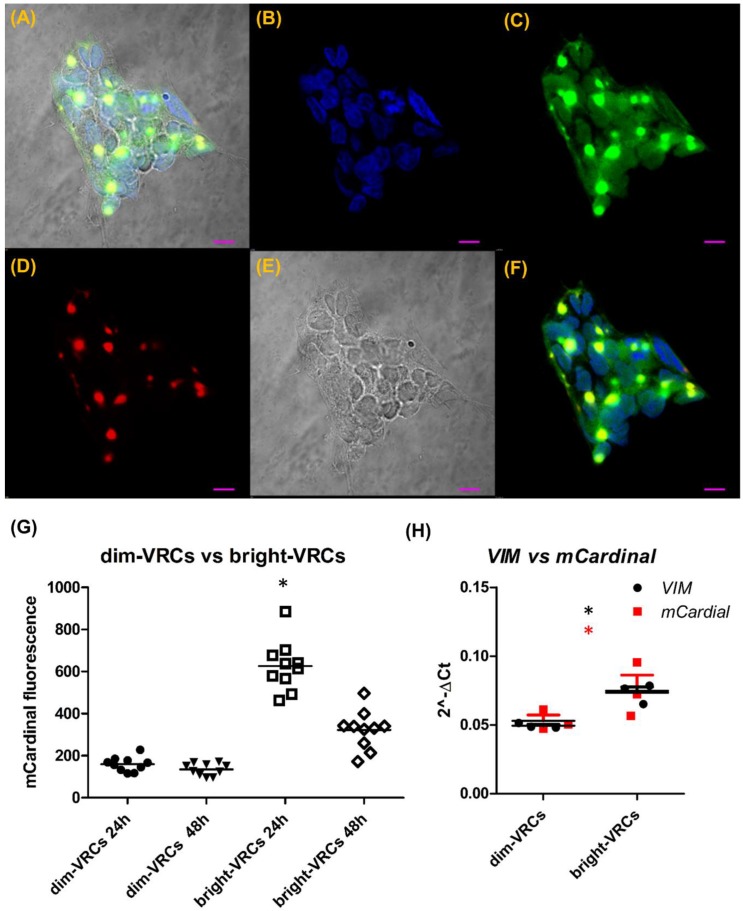
(**A**–**F**) Immunofluorescent labeling of FLAG-tagged mCardinal and VIM in VRCs. Shown in green, the FLAG tag (**C**) is seen as bright spot-like areas surrounded by less intensive fields localized in the cytoplasm. Localization of VIM shown in red (**D**) corresponds to the regions with the highest fluorescence of FLAG-tagged mCardinal (**C**). Blue colored DAPI, FLAG, VIM, and transmission are merged in picture **A**. (**B**) DAPI, (**C**) anti-FLAG, (**D**) VIM, (**E**) transmission, (**F**) merged fluorescence. The scale bar represents 10 µm. (**G**) mCardinal fluorescence changes of sorted VRCs populations with time. The fluorescence of dim-VRCs and bright-VRCs populations was measured in 24 h (dim-VRCs – filled circles, bright-VRCs – squares) and 48 h (dim-VRCs – filled inverted triangles, bright-VRCs – rhomboids) of culture upon sorting. The points correspond to the fluorescence of selected ROIs, whereas the lines show mean fluorescence. Statistically significant changes in fluorescence between treated vs. control cells are marked with an asterisk; * *p* ≤ 0.05 (Mann–Whitney test). (**H**) VIM (filled circles) and mCardinal (red filled squares) quantification by qPCR showed similar amounts of both transcripts. VIM and mCardinal genes were measured in dim-VRCs transfected cells in 24 or 48 h upon transfection. The data shows the mean 2^–ΔCT^ relative to GAPDH. The graph shows the representative result of the measurement, which was done in triplicate. The changes were statistically significant (* *p* ≤ 0.05) in comparison to the control group.

**Figure 3 cells-08-01644-f003:**
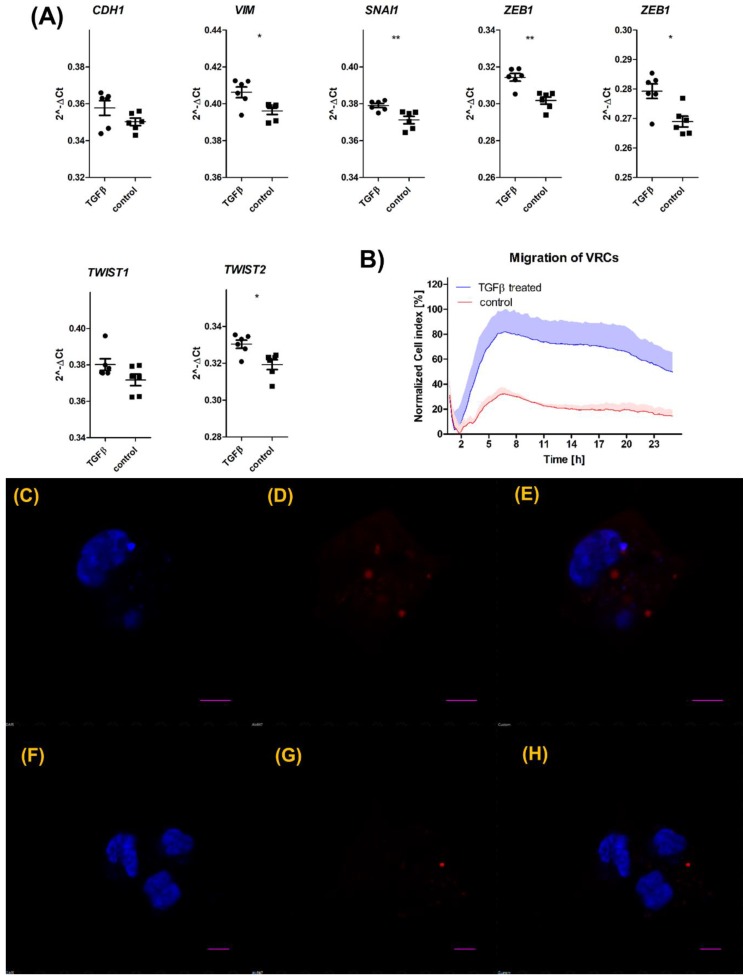
(**A**) *CDH1*, *VIM*, *SNAI1*, *ZEB1*, *ZEB2*, *TWIST1*, and *TWIST2* quantification by qPCR. Assessed genes were measured in VRCs untreated (squares) and treated (circles) with TGFβ for 72 h. The data shows mean 2^–ΔCT^ relative to *GAPDH*. The graph shows the representative result of the measurement. Statistical significance in comparison to control was calculated using the Mann–Whitney test and represented using the following annotations: * *p* ≤ 0.05; ** *p* ≤ 0.01. (**B**) Real-time migration analysis of transfected VRCs using xCELLigence system. The line shows mean normalized cell index, whereas the colored area depicts the standard deviation of three replicates. Migration of VRCs treated with TGFβ shown in blue, control shown in red. (**C**–**H**) Subcellular localization of mCardinal fluorescent protein in living VRCs treated with TGFβ for 72 h (**C**–**E**) and untreated (**F**–**H**). (**C**,**F**) nucleus (DAPI), (**D**,**G**) mCardinal, (**E**,**H**) merged. The scale bar represents 5 µm.

**Figure 4 cells-08-01644-f004:**
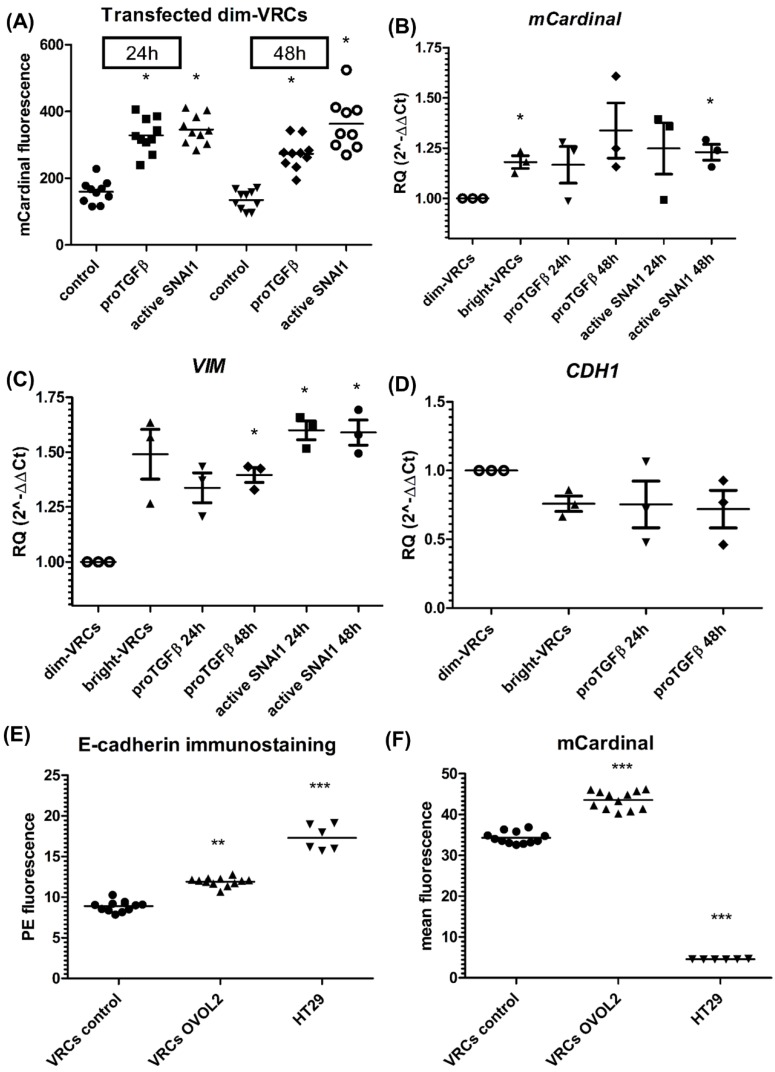
(**A**) mCardinal fluorescence of the dim-VRCs population transfected with active SNAI1 or proTGFβ plasmids. The fluorescence of the cells was measured after 24 h (control – filled circles, proTGFβ – filled squares, active SNAI1 – filled triangles) and 48 h (control – filled inverted triangles, proTGFβ – filled rhomboids, active SNAI1 – circles) of culture after sorting. The points correspond to the fluorescence of a selected ROI, whereas the lines show mean fluorescence. The fluorescence changes were considered statistically significant in comparison to the control group: * *p* ≤ 0.05 (Mann–Whitney test). (**B**) mCardinal and (**C**) VIM relative quantification (RQ) by qPCR. VIM and mCardinal genes were measured in transfected dim-VRCs 24 (proTGFβ – filled inverted triangles, active SNAI1 – filled squares) or 48 h (proTGFβ – filled rhomboids, active SNAI1 – filled circles) after transfection. dim-VRCs represented as circles, bright-VRCs represented as filled triangles. The data shows mean 2^–ΔΔCT^ relative to GAPDH. The results were normalized to the control, which was pUC18-transfected dim-VRCs. The graph shows the representative result of the measurement, which was performed in triplicate. (**D**) Relative quantification of CDH1 by qPCR. CDH1 levels were measured in proTGFβ transfected dim-VRCs 24 (proTGFβ – filled inverted triangles) or 48 h (proTGFβ – filled rhomboids) after transfection. The graph shows mean RQ 2^–ΔΔCT^ ± SEM relative to GAPDH. The results were normalized to the control, which was pUC18-transfected dim-VRCs 48 h after transfection. The graph shows the representative result of the measurement, which was performed in triplicate. dim-VRCs represented as circles, bright-VRCs represented as filled triangles. (**E**) Immunostaining of E-cadherin and (**F**) mCardinal fluorescence in VRCs and HT29 E-cadherin positive cells by flow cytometry. VRCs control—VRCs transfected with pUC18 vector; VRCs OVOL2—OVOL2-overexpressing cells 24 h after transfection. Immunostained untransfected HT29 cells were used as controls. The graph shows the single reads as well as the mean value of three independent experiments. Statistical significance in comparison to control group is represented by asterisks: * *p* ≤ 0.05; ** *p* ≤ 0.01; *** *p* ≤ 0.001. VRCs control represented as filled circles, VRCs OVOL2 represented as filled triangles, HT29 represented as filled inverted triangles.

**Figure 5 cells-08-01644-f005:**
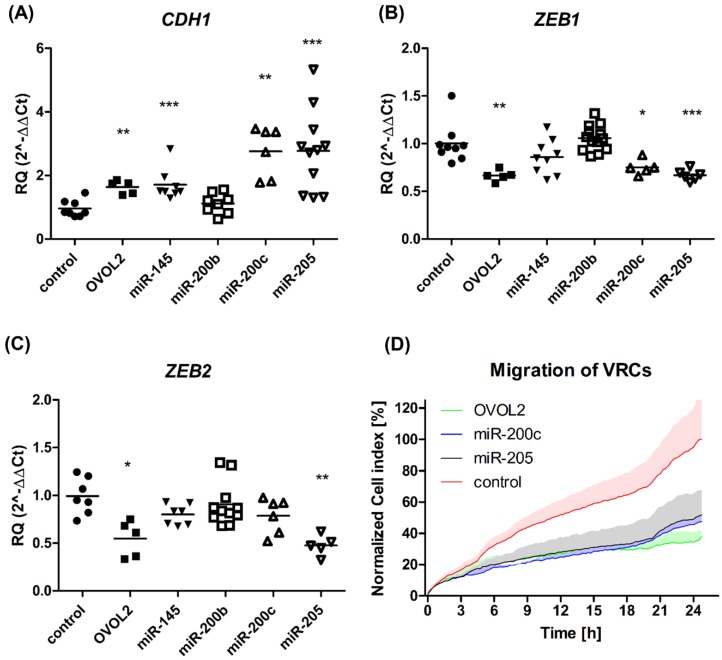
(**A**–**C**) Relative quantification of CDH1, ZEB1, and ZEB2 transcripts in transfected VRCs by qPCR. The expression of genes relative to GAPDH was measured 48 h after transfection of VRCs with OVOL2 (filled squares) and microRNA-expressing vectors (*miR-145 –* filled inverted triangles, *miR-200b –* squares, *miR-200c –* triangles, and *miR-205 –* inverted triangles). The results were normalized to VRCs control – filled circles (mock transfected) and shown as mean RQ 2^–ΔΔCT^ as well as single values. The graph contains data from at least two independent experiments, which were measured in triplicate. Statistical significance in comparison to control was calculated using the Mann–Whitney test and was rated by asterisk: * *p* ≤ 0.05; ** *p* ≤ 0.01; *** *p* ≤ 0.001. (**D**) Real-time migration analysis of transfected VRCs using xCELLigence system. The line shows mean normalized cell index, whereas the colored area depicts the standard deviation of three replicates. Migration of VRCs transfected with *OVOL2* is shown in green, *miR-200c* shown in blue, *miR-205* shown in black, whereas pUC18, the control, is colored in red. The representative results from two different experiments are presented.

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
