# Peer review of "Genetically Engineered Lung Cancer Cells for Analyzing Epithelial–Mesenchymal Transition"

_cells, 2019, doi:10.3390/cells8121644_

Round 1

Reviewer 1 Report

The concept of designing a VIM-reporter cell line is highly useful in the context of studying EMT in Health & Disease, and such tools are greatly appreciated by the scientific community. Do the authors are planning to share the cell model in the future to allow it wide use in the research labs? How are they planning to do?
Many control experiments are missing in the article, such as the level of expression of the genes that are controlling epithelial/mesenchymal phenotype of the cells. Moreover, it is surprising that the authors are not trying to simply directly treat the cells with growth factors to induce EMT. These experiments, that are basically easier than expressing plasmids, and closer to the biological context. They have to be added to the work.
The abnormal localization of Vimentin is a major concern for the model. And although it could be used to track cells with low VIM expression, this might impair the tumor cell plasticity and interfere with the study.
This is a very good work and the model need to be more explored to ensure its reliability and biological relevance. Overall the article is well written and easy to follow with a clear description of the works that have been done.

Author Response

We thank the reviewers and are grateful for their constructive comments. We have addressed the concerns raised during the reviewing of the manuscript and hereby resubmit the manuscript for re-evaluation. We feel that the manuscript has been significantly strengthened, and we hope that it would be found acceptable for publication.

All changes in the text are marked in red in the new version.

Do the authors are planning to share the cell model in the future to allow it wide use in the research labs? How are they planning to do?

We are going to share VRCs model for other researchers by deposing them into one or more of following human cell culture collections: American Type Culture Collection (ATCC),  The European Collection of Authenticated Cell Cultures (ECACC) and Leibniz Institute DSMZ German Collection of Microorganisms and Cell Cultures. We have already started the procedures required by ATCC. It seems that ATCC requires that the article is already published before acceptance of the line. Nevertheless, researchers can obtain the cell line by contacting the corresponding author.

Many control experiments are missing in the article, such as the level of expression of the genes that are controlling epithelial/mesenchymal phenotype of the cells. (...) Moreover, it is surprising that the authors are not trying to simply directly treat the cells with growth factors to induce EMT. These experiments, that are basically easier than expressing plasmids, and closer to the biological context. They have to be added to the work.

The VRCs have always been compared to the parental line or to themselves e.g. before and after EMT/MET stimulation. In all cases the experiments were done in at least triplicates and repeated at least twice. In the previous version, we have used well-known EMT drivers, genetically coded (proTGF or SNAI1), and showed that these molecules are able to induce EMT. Now, we have also used pure TGFbeta as a EMT inducer, and in this way we reveal that the transcripts of classical mesenchymal genes such as VIM, SNAI1, ZEB1, ZEB2, TWIST1, and TWIST2 were elevated in VRCs treated with TGFbeta (10 ng/mL) for 72 hours in comparison to untreated cells. The new data is presented in one new Figure 3.

The abnormal localization of Vimentin is a major concern for the model. And although it could be used to track cells with low VIM expression, this might impair the tumor cell plasticity and interfere with the study.

We agree with the reviewer that this might be a concern for some phenotypical responses of this line. Nevertheless, the VRCs behave as expected when MET is induced (miRNAs or OVOL2) by increasing E-cad and reducing VIM expression and migration. We have further tested TGFb-induced EMT on VCRs, and as expected the EMT signature is present (see above) as well as a significant promotion of cell migration (Figure 3B). The Vimentin network still exists as it can be observed in the confocal images by immunofluorecent labeling of Vimentin.

Reviewer 2 Report

In this manuscript, Kielbus M et al. established reporter lung cancer cell line which integrated fluorescence protein (T2A-mCardinal) into VIMENTIN (VIM) gene locus for detecting EMT, MET and hybE/M. Data is good and this cell line will be good material for EMT studies, but two points need to be improved.

Major point
In Figure 2A, please show representative confocal microscopy images of VIM-T2A-mCardinal localization in each condition (Control, proTGFb and activeSNAI1).

Minor points
They use “localisation, or localised” and “localization” as same meanings. It may be better to use “localization”. Please check and correct them.

Author Response

We thank the reviewers and are grateful for their constructive comments. We have addressed the concerns raised during the reviewing of the manuscript and hereby resubmit the manuscript for re-evaluation. We feel that the manuscript has been significantly strengthened, and we hope that it would be found acceptable for publication.

All changes in the text are marked in red in the new version.

Major point
In Figure 2A, please show representative confocal microscopy images of VIM-T2A-mCardinal localization in each condition (Control, proTGFb and activeSNAI1).

While mCardinal, probably as uncleaved Vim-mCardinal, is mostly concentrated in foci, the Vimentin network still exists (immunolabelling of Vimentin in Figure 2), showing that P2A processing works to some degree. While only brightness changes, as we expected, during EMT and becomes dimmer during MET, the localisation of mCardinal is unchanged.

Minor points
They use “localisation, or localised” and “localization” as same meanings. It may be better to use “localization”. Please check and correct them.

Thanks for noticing this, we have changed the British to US spelling throughout the text.

Reviewer 3 Report

In this work, the authors engineered and characterized lung cancer cell by  knocking-in the fluorescent protein coding gene (mCardinal). Many experiments were done and illustrated that VRCs can be a reliable reporter model for studying EMT and MET. To my impression, the manuscript is presented in a well-organized and logical manner. All the experimental results obtained from their studies show reasonable consistency. In addition, these studies will contribute to further studies on its applications. I would therefore strongly recommend this manuscript for publication Cells.

Author Response

We thank the reviewer for the encouraging comments.

We have expanded the manuscript according to the comments of all 3 reviewers. A new version has been submitted for consideration.

Round 2

Reviewer 1 Report

The changes significantly increased the quality of the manuscript.

However, statistical analysis are not clearly displayed in most of the figures. Including on the new figures. 

This need to be fixed to evaluate the strength of the reporter.

Author Response

We would like to thank the reviewer for noticing this. Somehow we missed marking the statistics within the Figures. This has been corrected in the new version.